# Comparative Study on Water Vapour Resistance of Poly(lactic acid) Films Prepared by Blending, Filling and Surface Deposit

**DOI:** 10.3390/membranes11120915

**Published:** 2021-11-23

**Authors:** Shuo Wang, Qinyu Shen, Chuanyan Guo, Hongge Guo

**Affiliations:** School of Light Industry Science and Engineering, Qilu University of Technology, Jinan 250353, China; 10431200886@stu.qlu.edu.cn (S.W.); 10431210997@stu.qlu.edu.cn (Q.S.); 10431210995@stu.qlu.edu.cn (C.G.)

**Keywords:** polylactic acid blown film, modification, bio-oriented PLA, atomic layer deposition, water vapour transmission rate

## Abstract

The polylactic acid (PLA) resin Ingeo 4032D was selected as the research object, with a focus on PLA modification by using polymers such as linear low-density polyethylene (LLDPE), high-density polyethylene (HDPE) and ethylene–propylene-diene monomer grafted with glycidyl methacrylate (EPDM-g-GMA), by using fillers such as nano calcium carbonate and zeolite. In order to characterize the deposition effect of Al_2_O_3_ on the film surface by plasma-assisted atomic layer deposition, Bio-oriented PLA (BOPLA) with more uniform thickness than blown film was purchased for study. The mechanical properties, friction coefficient, surface contact angle and water vapour transmission rate of the modified PLA film were compared and discussed. The aim was to find out the most influencing factors of film’s water vapour resistance.

## 1. Introduction

PLA resin is a biodegradable polymer material prepared by polycondensation reaction using lactic acid as the main raw material, which can be extracted from natural renewable resources such as corn and wheat [1,2,3]. The production process has no pollution and the production energy consumption is only equivalent to 20% to 50% of the traditional petrochemical products, while the carbon dioxide gas produced is only 50%, which can get rid of the dependence on oil resources. After being discarded, PLA products can be decomposed in the natural environment and eventually degraded into water and carbon dioxide [4], without any pollution to the environment and realizing its circulation in the natural world. The molecular chain structure of PLA is shown in Figure 1. It is considered to be the most widely used type of biodegradable polymer material and has received wide attention in recent years.

PLA resin is a thermoplastic aliphatic polyester with a glass transition temperature between 55 and 65 °C, whose melting point is about 175 °C and whose processing temperature is generally controlled between 170 °C and 230 °C, which is suitable for extrusion, blow molding, stretching and other processing processes [5].

PLA film has high crystal transparency, stable physical properties such as good solvent resistance and insolubility in alcohols, fats, hydrocarbons and edible oils. Moreover, there is a lower temperature heat sealing ability, better printability than polyolen films and good ink retention, which can retain the flavor and package aesthetics of foods to a greater extent. It makes PLA resin a good choice for replacing fossil-based plastics generally used in food packaging industry [1,6,7,8,9].

PLA resin also inevitably has certain drawbacks, including its low gas barrier properties compared to high-performing conventional plastics, its low thermal stability and its high price still restrict its use in the market [1,9,10,11,12], and these deficiencies needs to be overcome if PLA is to be successfully used in packaging or other applications requiring good barrier performance.

A barrier is an important standard to measure packaging materials such as moisture permeability (WVTR) and oxygen permeability (OTR). In recent decades, many researchers have proposed a variety of feasible methods to improve the barrier performance of polylactic acid. There are usually three physical methods: polymer blending modification, polymer filling modification and surface coating of film.

For polymer blending modification, Sun et al. [13] prepared the films by blending poly(ε-caprolactone) (PCL) and PLA, and the barrier performance of the films were improved by controlling the PCL content. Yu et al. [14] improved the barrier properties of PLA by incorporating of nanocelluloses (NCs) and focused on the influence of the differences in NC morphology and dimensions on the PLA properties. This comparative study was very beneficial for selecting reasonable nanocelluloses as nucleation/reinforcing agents in robust-barrier packaging biomaterials with outstanding mechanical and thermal performance. Le Gars et al. [6] also filled NCs into PLA; in contrast, the nanocellulose acts as an inner layer between the two PLA films to enhance the oxygen barrier in their research.

The blending of PLA with immiscible organic compounds such as polyethylene (PE) and polypropylene (PP) has also become a common way to improve the properties of polylactic acid but requires compatibilizers to achieve stable morphologies and superior mechanical properties. Thurber et al. [15] showed that catalyst localized at the interface can compatibilize polyethylene (PE) and polylactide (PLA) blends. In this work, compatibilized blends of HDPE (high-density PE) and PLA are created using modest amounts of hydroxyl functional PE. The molecular chain structure of PE is shown in Figure 2. This branching of PE would give an impact on the entanglement of PE with PLA, which would change the mechanical strength of blending film.

In Xu et al. [16], polylactide (PLA) was melt-blended with either polypropylene (PP). An ethylene−glycidyl methacrylate−methyl acrylate terpolymer (Lotader) was utilized as compatibilizer through coupling to the end groups of PLA. The remarkable efficacy of PEGMMA as a reactive compatibilizing agent allows the bridging of two immiscible but important classes of thermoplastics, polylactide and polypropylene, and the production of ductile PLA/PP blend materials.

PP, PE and other auxiliaries are added to the raw materials to achieve biodegradation. However, this view is doubtful; some people call it “pseudo-degradation”, adding auxiliaries can only allow the plastic to be degraded into particles invisible to the naked eye, and its harm to the environment still exists and cannot be completely biodegraded. At present, the main purpose is to reduce the cost.

For polymer filling modification, Jong et al. [17] fabricated PLA-based composite films with different types of nano-clays as strengthening agent by a solvent casting method. They found that the water resistance of PLA films improved to different degrees. In order to keep the film’s flexibility, the selection of plasticizer is also important. Giuseppe Mele et al. [18] combined cardanol oil (CA) with PLA and prepared PLA/CA films by means of hot melt extrusion processes. The presence of CA increased the oxygen transmission through the PLA/CA films; consequently, the permeability values were always appreciably higher for plasticized films. Nevertheless, the CA-plasticized PLA films showed good barrier properties similar to packaging materials commonly used in the food industry today.

For surface coating of film, Satam et al. [19] found that bioavailable cellulose nanocrystals (CNC), chitin nanofibers (ChNF), are readily dispersed in water, enabling spray-coated films to be deposited at high rates onto uneven or delicate surfaces. They sprayed coating of cationic chitin nanofibers and anionic CNC suspensions onto PLA films layer-by-layer to improve the oxygen barrier. ChNF/CNC multilayers were found to lead to a reduction in the O2 permeability of the final composite film by as much as 73% with the largest effects seen in composites with three alternating layers (ChNF-CNC-ChNF). Multilayer ChNF/CNC coatings were found to have lower O_2_ permeability and lower haze than those coated with ChNF or CNCs alone (72% and 86% lower haze, respectively), pointing to a synergistic effect. The composites had a water vapour transmission rate similar to the PLA substrate. Mericer et al. [20] prepared multilayer composite films by coating incompatible microfibrillated cellulose on amorphous polylactic acid and semicrystalline polylactic acid, respectively. The results showed that the oxygen barrier of multilayer films was improved by more than one order of magnitude. Mattioli et al. [21] deposited hydrogenated amorphous carbon on the surface of PLA thin films by CVD to improve the barrier properties of PLA thin films. They found that the moisture and oxygen permeability of hydrogenated amorphous carbon/PLA films decreased significantly when the treatment time was 5 min. Hirvikorpi et al. [22] deposited Al_2_O_3_ with different thickness onto 40 μm thick PLA film with a roll-to-roll process. The results show that the PLA/Al_2_O_3_ structure significantly improves the gas barrier of the film.

Our team’s research starts from three aspects. Firstly, a small amount of synthetic resin used for packaging film (LLDPE and HDPE) are selected to modify PLA ‘s brittleness without reducing its rigidity. Due to the different polarity between PLA and PE, there is a compatibility problem; therefore, GMA-grafted ethylene propylene copolymer (EPDM-g-GMA) is selected to improve the toughening effect. Secondly, PLA was modified by filling with nano calcium carbonate, zeolite and plasticizing with Epoxidized soybean oil (ESO). Finally, for the anisotropy and non-uniformity of surface thickness of blown film, BOPLA for tape casting was selected for depositing Al_2_O_3_ on it. The mechanical, surface and water vapour barrier properties of PLA film by three modification methods were summarized. The experimental results have reference value for the production requirements of food and drug packaging film.

## 2. Experimental

### 2.1. Materials

PLA (Ingeo 4032D, Degree of polymerization:200) was purchased from Nature Works Company (Blair, NE, USA). At a temperature of 190 °C, loading pressure of 2.16 kg and cutting time interval of 30 s, the MFR of Ingeo 4032D PLA was 5.872 g/10 min.

BOPLA film (40 μm) was purchased from Shandong Shenghe Plastic Development Company (Weifang, China).

LLDPE (7042, $\bar{M}$_w_: 2.71 × 10^4^) was purchased from China Petrochemical Corp. (Beijing, China).

HDPE (DMDA8007, $\bar{M}$_w_: 7.36 × 10^4^) was purchased from Shenhua Energy Company (Beijing, China).

EPDM-g-GMA (E533, grafting rate: 1%,) was purchased from Ningbo New Material Corp. (Ningbo, China). The molecular chain structure of EPDM-g-GMA and GMA (Glycidyl methacrylate) is shown in Figure 3.

Nano-CaCO_3_ (Nanometer scale calcium carbonate, with an average particle size of 10–100 nm) was purchased from Haofu Chemical Co., Ltd. (Shanghai, China).

Zeolite (3A molecular sieve, with an average particle size of 2–5 µm) was purchased from Yuanli Chemical Co., Ltd. (Tianjin, China).

Epoxidized soybean oil (ESO) was purchased from Xinjinlong Plastic Additives Co., Ltd. (Guangzhou, China).

Trimethyl aluminum (TMA, purity: 99.99%) was purchased from Sigma-Aldrich Trading Co., Ltd. (Shanghai, China).

### 2.2. Film Manufacting and Depositing Methods

Pretreatment: PLA resin was dried in vacuum drying oven at 80 °C for 4 h.

Mixed granulation: Raw materials were mixed in different proportions, and the granular resin for blowing film was produced by the extruding part of torque rheometer.

Film sample preparation: The granular resin was then added to Blown-Film part of torque rheometer. The processing temperatures of each heating area are 155 °C, 170 °C, 175 °C, and 185 °C, the rotating speed is 35 rpm, the blow-up ratio is 2.6 and the draw ratio is 5.0. May vary slightly according to different materials.

Surface deposition: Plasma-assisted atomic layer deposition is carried out by input TMA using a self-designed equipment under atmospheric pressure (Figure 4). The deposition temperature is 80 °C.

### 2.3. Performance Testing and Characterization Methods

#### 2.3.1. Mechanical Property Test

According to the plastic tensile test method of GB/T1040.2-2006, the longitudinal tensile strength of each sample was tested by intelligent electronic tension tester (XLW (EC)), which made by Labthink Electromechanical Technology company of China. The size of sample is 100 mm × 15 mm, and the tensile speed is 200 mm/min. Each sample was tested more than 5 times and the average value was taken.

#### 2.3.2. Friction Coefficient Test

According to the method of measuring the friction coefficient of plastic film and sheet by GB/T10006-1988, Friction coefficient meter (MXD-01) made by Labthink Electromechanical Technology company of China was used to measure the friction property. Two samples with specifications of 200 mm × 80 mm and 63 mm × 63 mm were cut and moved relatively at a consistent speed. The force values were recorded and calculated. Each sample was tested more than 5 times, and the average value is taken. For the calculation method of friction coefficient, see Formulas (1) and (2):(1)$$\mu_{s}\;=\;\frac{F_{s}}{F_{p}}$$
where *μ**_s_* is the coefficient of static friction, *F**_s_* is the static friction with N as units, *F**_p_* is the normal force with N as units.
(2)$$\mu_{d}\;=\;\frac{F_{d}}{F_{p}}$$
where *μ**_d_* is the dynamic friction coefficient, *F**_d_* is the dynamic friction with N as units, *F**_p_* is the normal force with N as units.

#### 2.3.3. Contact Angle Test

The standard Deionization Water Drops were added to the surface of the tested sample with a micro sampler to form drops. We waited for 3 min until the drops were stable and then measured the contact angle between the sample and water. Static drop contact angle measuring instrument (DSA100) made by Kruss company of Germany was used to measure the contact angle of pure water on PLA film. Each sample was tested more than 5 times and the average value was taken.

#### 2.3.4. Moisture Barrier Property Test

A: According to the test method for water vapour transmission of plastic film by GB/T 30412-2013, moisture permeability instrument (PERMATRAN-W 1/50G) made by MOCON company of USA was used to measure the moisture permeability on modified PLA film. The working principle is shown in Figure 5. The size of sample is 50 cm^2^ and the experimental temperature is 38 °C. Each sample was tested more than 5 times and the average value was taken.

B: According to the test method for water vapour transmission of plastic film (Cup method) by GB/T 1037-1988, constant temperature and humidity testing machine (GT-7005-T) made by the Gotech testing instruments (Dongguan) company of China was used to measure the moisture permeability on PLA film. The sample is a disc with a diameter of about 40 mm. The experimental temperature and relative humidity were 38 °C and 90%, respectively. The moisture permeable cup needed for the experiment is shown in Figure 6. Each sample was tested more than 5 times and the average value was taken. The permeability is calculated using Equation (3) below,
(3)$$Q\;=\;\frac{24\;\times \;\Delta m}{t\;\times \;A}$$
where *Q* is the Water Vapour Transmission Rate (WVTR), the unit is g/(m^2^·24 h), ∆m is the amount of vapour passing through the film per unit time, the unit is g, t is per unit time, and A is the effective area of sample; the unit is m^2^.

## 3. Results and Discussion

### 3.1. The Mechanical Properties of Modified PLA Films

PLA resin, as a self-degrading biopolymer, has many advantages, but it still cannot fully meet all the requirements for packaging. It is an aliphatic polymer with a loose helical molecular chain [23]. The longitudinal tensile strength of its blown film is nearly twice that of the transverse strength. It is hard, brittle and low in crystallinity. It can easily absorb moisture in a humid environment, resulting in film swelling, and it has poor a barrier to water vapour. However, its biaxial tension film by tape casting method is highly oriented and crystalline with uniform high strength and elongation, as shown in Table 1.

Linear Low-Density Polyethylene (LLDPE), High Density Polyethylene (HDPE) or EPDM rubber grafted glycidyl methacrylate (EPDM-g-GMA) are selected for PLA blend modification. LLDPE is ethylene with a small amount of α- Olefin copolymerization in the presence of catalyst between the copolymer; it has a linear ethylene main chain with a very short molecular structure of comonomer branches and has good strength, toughness and rigidity. HDPE is linear and has a long molecular chain with a molecular weight of hundreds of thousands. EPDM-g-GMA is an elastomer, EPDM is a copolymer of ethylene, propylene and a small amount of non-conjugated dienes. GMA is a polar graft group with good compatibility with PLA. Nano calcium carbonate (nano-CaCO_3_) and zeolite were used as fillers, and ESO was used as plasticizer. All these three polymers and additives meet the hygienic requirements for the use of food and drugs. At first, the mixture was granulated in an extruder and then blown to obtain various films.

From Figure 7, the longitudinal tensile strength of the blend films are higher than that of pure PLA film. The result of adding LLDPE has the best tensile strength and EPDM-g-GMA has the best elongation. This means that a small amount of polyolefin has good compatibility with PLA and has no much difference on the rigidity of PLA itself and EPDM-g-GMA has indeed played a toughening effect.

From Figure 8, compared with PLA resin and its blends with HDPE or HDPE together with LLDPE, the addition of 2 phr of nano-CaCO_3_ really plays a reinforcing role and improves the strength slightly but cannot improve the toughness of the blends.

From Figure 9, adding ESO as plasticizer can improve the toughness of PLA film, but significantly reduce the strength of the material. As a filler, Nano-CaCO_3_ works better than zeolite because the particle size of zeolite is relatively large at an average particle size of 2–5 µm.

Plasma-assisted atomic layer deposition technique was used to deposit Al_2_O_3_ thin films on extruded bio-oriented PLA film of 40 μm [24]. Seen from Figure 10, with the increase in deposition cycles, the tensile strength of both vertical and horizontal direction of the films increased significantly and the elongation had not changed much.

The above three modification methods do not significantly reduce the mechanical properties of PLA as packaging film; the influence of modification on the surface properties of the film is discussed below.

### 3.2. The Surface Properties of Modified PLA Films

The coefficient of friction is a measure of the friction between two contact surfaces. In the micro world, the material surface is uneven. When the two materials are in contact with each other, only the convex parts are in real contact, and the atoms in the convex parts are in close contact to form a strong interaction force. When the contact surfaces moves relative to each other, this force will be hard shear. The convex parts of the two contact surfaces collide with each other and break and wear, forming an obstacle to the movement of the object. The shear force parallel to the contact surface and destroying the convex part is friction. Adhesion is an adhesion phenomenon between plastic film contact layers, which is usually caused by two situations: the extremely smooth film surface is in close contact and almost completely isolated from the air; the pressure and temperature cause the adhesion and melting of the film contact surface. Adhesion will make the friction coefficient show an upward trend.

When used as packaging film, the inner and outer surfaces of the film shall have good smoothness to ensure smooth filling on the high-speed automatic filling machine. Therefore, the dynamic and static friction coefficient of the film surface shall be relatively low, generally from 0.2 to 0.4 [25].

As seen from Table 2, the dynamic and static friction coefficients on the inner side of all blown films are greater than those on the outer side. This is because in the film-blowing process, the film forms a closed cylindrical shape of closed compressed air, the temperature of the inner surface film higher than that of the outer surface. The reason for this phenomenon is that the crystallization speed of PLA is relatively slow. There was a wind ring on the outside during the film blowing process, the film was quenched, and the crystal was generated to a lesser extent. The inner side of the film was hot air, the molecular chain can be arrayed into lattice before cooling, and the crystal was generated. Therefore, the inner surface is rough and the friction coefficient is larger than that of the outer surface.

As shown in Table 2, compared with pure PLA, the friction coefficient of the blend film is lower than the original, indicating that the packaging opening of the blend film is improved. Table 3 shows that, after adding Nano-CaCO_3_, the friction coefficient of the film increases because the dispersion of Nano-CaCO_3_ in PLA is not so good that many Nano-CaCO_3_ particles migrate to the material surface to increase the surface roughness. As shown in Table 4, after adding ESO, the friction coefficient of the blend membrane decreased. ESO increases the compatibility between the filler and PLA and acts as a lubricant. However, the amount of ESO should not be too much; otherwise it will cause adhesion on the film surface and increase the friction coefficient. Compared between Nano-CaCO_3_ and Zeolite, because the size of Nano-CaCO_3_ (10–100 nm) is smaller that of Zeolite (2–5 µm) and the dispersion is uniform, the friction coefficients are smaller.

The surface contact angle is another method to characterize the surface properties of thin films. Plasma-assisted atomic layer deposits Al_2_O_3_ on PLA film, the surface properties of the base film affect the deposition effect (Figure 11). When polar groups are introduced into the surface, the smaller the water contact angle is, the more conducive it is to deposition. However, controlling the dense, uniform and ultra-thin deposit on PLA surfaces remains a challenge.

### 3.3. The Water Vapour Resistance of Modified PLA Films

Packaging is the most important way to protect commodities, especially food. In order to make packaging materials used in the circulation of commodities, we must improve their barrier to gas, especially water vapour. Barrier performance is not only an important evaluation standard to measure packaging materials, but also one of the main bases by which to predict the shelf life of products. The barrier refers to the performance of materials that hinder the penetration of small molecule gases, water vapour and organic solution vapour. Among them, moisture permeability is the main parameter to judge the water-vapour resistance of the film. When there is a pressure difference between the water vapour on both sides of the film, the water vapour will expand from the high-humidity side to the other, as seen in Figure 12. In this study, the film was prepared by a speed of 35 rpm at 155–185 °C, which suggested the possibility that the disentanglement rate > shear rate determined the crystallinity of films [26]. This crystallinity of film might determine the water vapour permeability shown in Figure 12.

Two kinds of methods were used to measure the Water Vapour Transmission Rate (WVTR) by a moisture-permeable cup or moisture permeability tester. The former is the weight gain method, and the latter is the weight loss method. The surface roughness, hydrophilicity, thickness and thickness uniformity of the film all affect the test results. The inner surface of the blown film was the high humidity side. The WVTR test range of the instrument (Mocon PERMATRAN-W 1/50G) is 0.1–100 g/m^2^, 24 h. PLA blown film’s WVTR is too high to test it by, so a moisture-permeable cup method was used for blown films. As shown in Table 5, when blended with general plastics, elastomer filler with plasticizer significantly reduces WVTR of films; however, it cannot meet the requirements of moisture barrier materials. The shape, particle size and dispersion of filler, dosage, adhesion to matrix and porosity all affect WVTR. Another reason that film of PLA/ESO/nano-CaCO3 (100:3:4 wt%) has the best effective barrier effect on water vapour may be that nano-CaCO3 is well dispersed in PLA, and as a nucleating agent, the film has higher crystallinity.

Compared with the non-stretched film, the mechanical properties and barrier properties of film produced by bi-axial stretching technology are significantly improved. BOPLA film was used as deposited base film.

In order to meet the requirements of moisture resistance under harsh conditions, atomic layer deposition is an effective method to prepare high barrier films [27]. Compared with the traditional lamination technology, it has the advantages of obviously saving the weight of packaging materials, significantly reducing production costs, and effectively avoiding environmental pollution.

Table 6 shows that it has a significant effect on improving the water vapour barrier performance of Al_2_O_3_ films, because dense Al_2_O_3_ has strong inter-atomic and inter-molecular bonds, as well as short-range ions and covalent bonds, leaving little space between atoms for the dissolution and diffusion of water vapour.

## 4. Conclusions

Blending, filling, both filling and plasticizing, and depositing Al_2_O_3_ all can keep or increase the longitudinal tensile strength of modified film, but the improvement of elongation is not obvious. The surface properties characterized by the friction coefficient show that most of the film meets the requirements of high-speed filling. Although filling reinforcement can improve the performance of packaging materials, food contact materials containing nano materials and plasticizers may cause safety problems, such as the inherent toxicity of materials and the risk of migration from materials to food and possible ingestion by consumers. Stretching oriented technology and plasma-assisted atomic layer deposition of Al_2_O_3_ is an effective tool to enhance the barrier performance of PLA and expand its application in the future. Surface deposition can significantly reduce WVTR of PLA, with a reduction range from 79.57% to 99.36%, while other methods can reduce it to a small extent, with a reduction range from 37.11% to 74.30%.

However, the preparation of ultra-thin solid Al_2_O_3_ barrier layer on PLA surface is still challenged by the very low glass transition temperature of PLA. In addition, whether the deposition of Al_2_O_3_ will affect the optical and degradation properties of PLA films is also one of the problems to be considered.

## Figures and Tables

**Figure 1 membranes-11-00915-f001:**
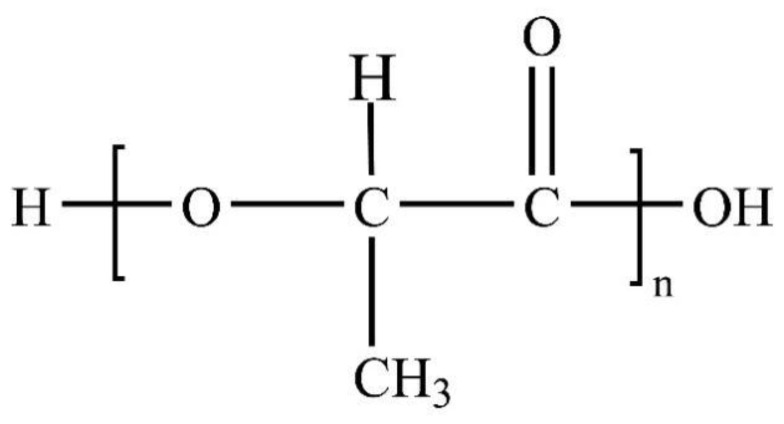
Molecular chain structure of PLA.

**Figure 2 membranes-11-00915-f002:**
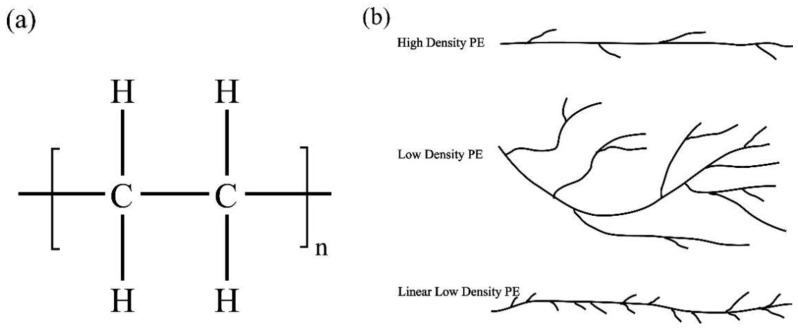
Molecular chain structure (**a**) and schematic diagram (**b**) of PE.

**Figure 3 membranes-11-00915-f003:**
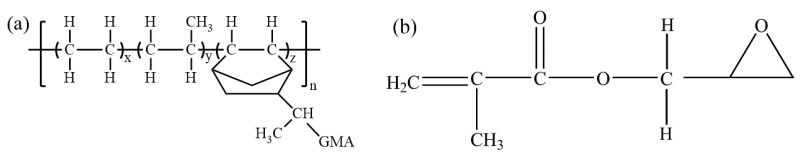
Molecular chain structure of EPDM-g-GMA (**a**) and GMA (**b**).

**Figure 4 membranes-11-00915-f004:**
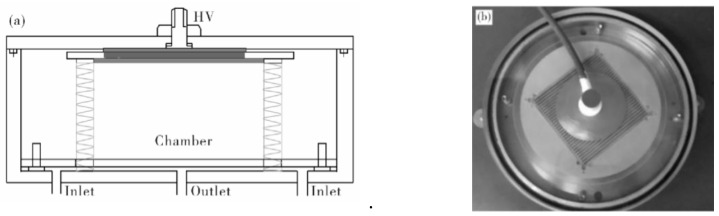
(**a**) The schematic diagram and (**b**) setup photo of equipment.

**Figure 5 membranes-11-00915-f005:**
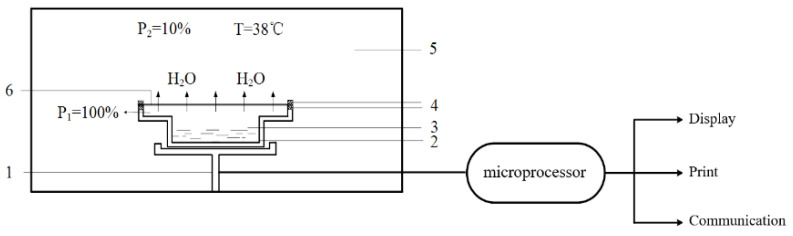
Schematic diagram of measuring principle of moisture permeable tester, where 1: Weighing sensor; 2: Moisture permeable cup; 3: Deionization water; 4: Sealing ring; 5: Constant temperature and humidity environment; 6: Film.

**Figure 6 membranes-11-00915-f006:**
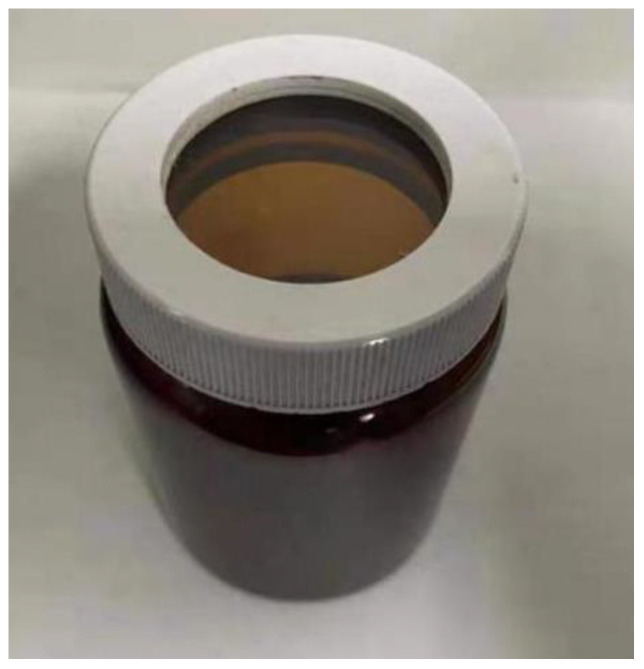
Moisture permeable cup.

**Figure 7 membranes-11-00915-f007:**
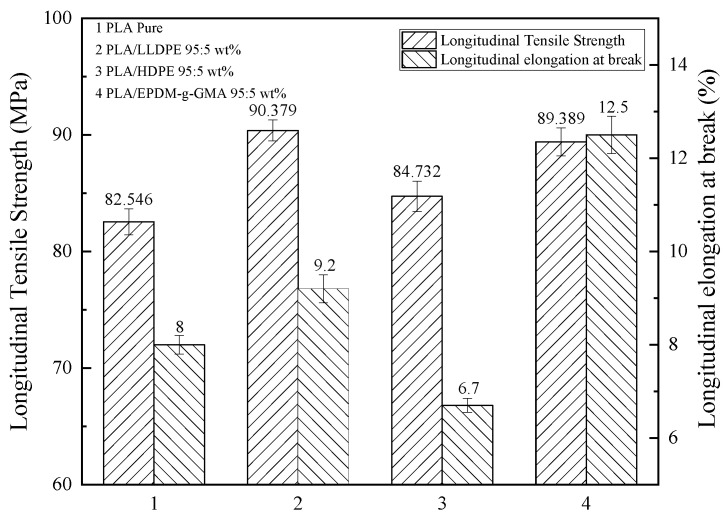
The longitudinal tensile strength and longitudinal elongation at break of PLA blends (film thickness: 30–80 µm, tensile speed: 200 mm/min).

**Figure 8 membranes-11-00915-f008:**
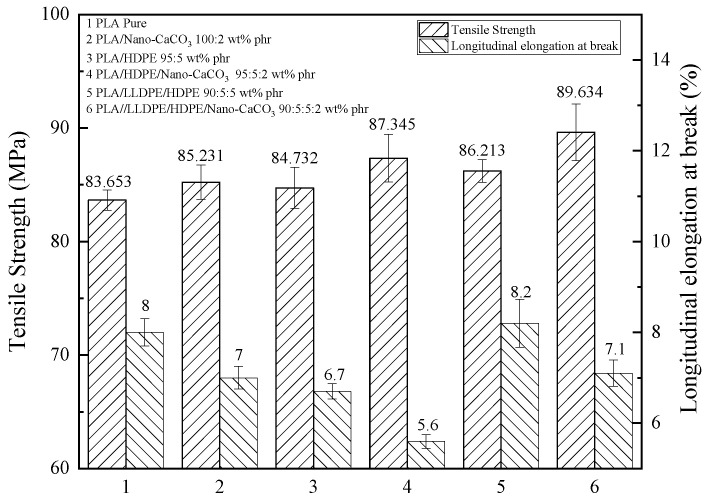
The longitudinal tensile strength and longitudinal elongation at break of filled PLA (film thickness: 30–80 µm, tensile speed: 200 mm/min).

**Figure 9 membranes-11-00915-f009:**
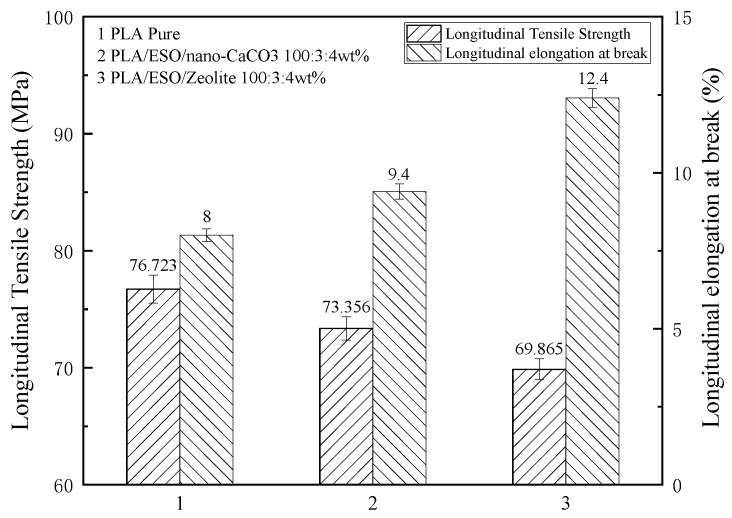
The longitudinal tensile strength and longitudinal elongation at break of filled and plasticized PLA (film thickness: 30–80 µm, tensile speed: 200 mm/min).

**Figure 10 membranes-11-00915-f010:**
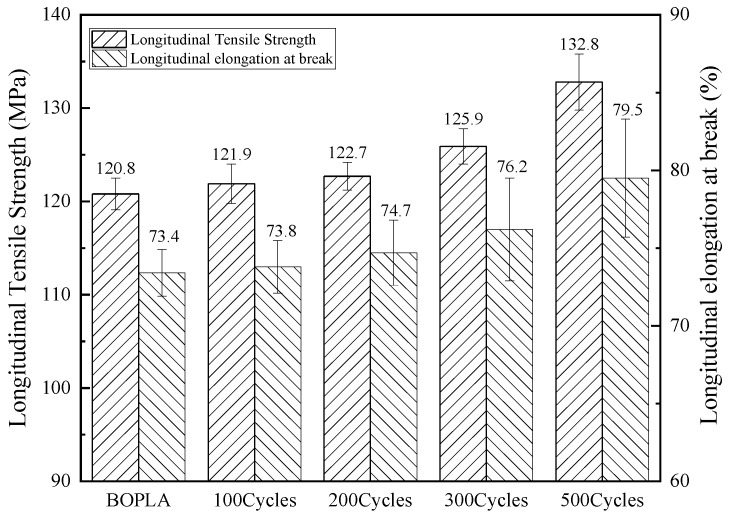
The longitudinal tensile strength and longitudinal elongation at break of Al_2_O_3_-deposited BOPLA film (tensile speed: 200 mm/min).

**Figure 11 membranes-11-00915-f011:**
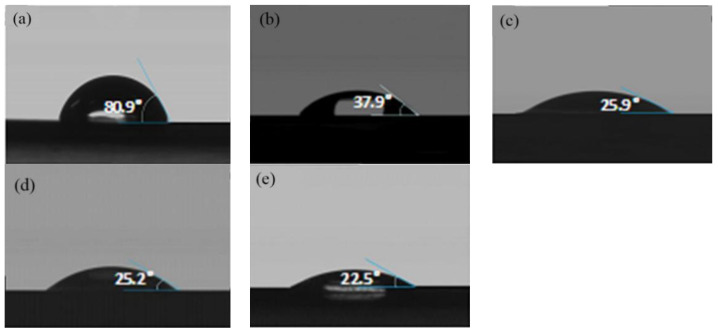
(**a**–**e**) Contact angle change of untreated material, after oxygen plasma treatment 1 min, 3 min, 5 min and 10 min.

**Figure 12 membranes-11-00915-f012:**
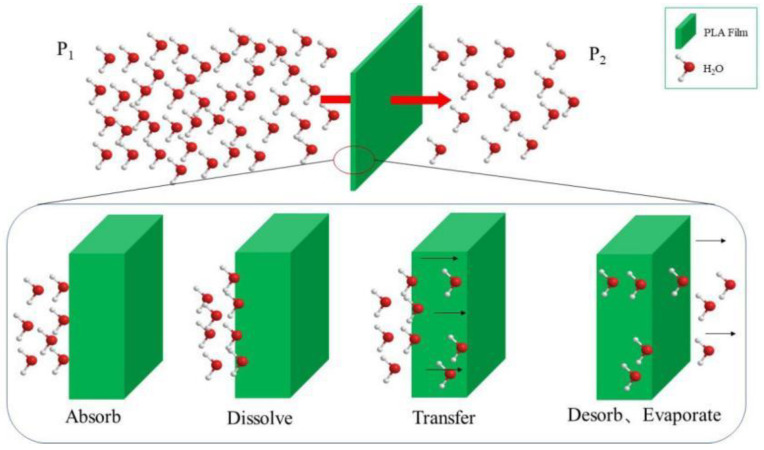
The expand process of water vapour from high humidity side to the other side.

**Table 1 membranes-11-00915-t001:** Mechanical properties of pure PLA blown film and bio-oriented PLA film.

Film Material	Thickness (μm)	Longitudinal Tensile Strength (MPa)	Longitudinal Elongation at Break (%)	Transverse Tensile Strength (MPa)	Transverse Elongation at Break (%)
Pure PLA blown film	35 ± 15	82.546	8.0	48.819	5.7
Bio-oriented PLA film	40 ± 5	120.8	73.4	138.1	122.1

**Table 2 membranes-11-00915-t002:** Friction Coefficient Data of PLA blends.

Direction	Longitudinal	Inner Side	Outer Side
Pure PLA	Static	0.503	0.470
	Dynamic	0.466	0.444
PLA/LLDPE (95:5 wt%)	Static	0.593	0.302
	Dynamic	0.531	0.211
PLA/HDPE (95:5 wt%)	Static	0.553	0.330
	Dynamic	0.499	0.292
PLA/EPDM-g-GMA (95:5 wt%)	Static	0.337	0.315
	Dynamic	0.330	0.316

**Table 3 membranes-11-00915-t003:** Friction Coefficient Data of filled PLA.

Direction	Longitudinal	Inner Side	Outer Side
Pure PLA	Static	0.503	0.470
	Dynamic	0.466	0.444
PLA/Nano-CaCO_3_ (100:2 wt%)	Static	0.593	0.302
	Dynamic	0.531	0.211
PLA/HDPE (95:5 wt%)	Static	0.553	0.330
	Dynamic	0.499	0.292
PLA/HDPE/LLDPE (90:5:5 wt%)	Static	0.490	0.359
	Dynamic	0.459	0.328
PLA/HDPE/Nano-CaCO_3_ (95:5:2 wt%)	Static	0.623	0.327
	Dynamic	0.558	0.232
PLA/HDPE/LLDPE/Nano-CaCO_3_ (90:5:5:2 wt%)	Static	0.605	0.275
	Dynamic	0.529	0.224

**Table 4 membranes-11-00915-t004:** Friction Coefficient Data of filled and plasticized PLA.

Direction	Longitudinal	Inner Side	Outer Side
Pure PLA	Static	0.503	0.470
	Dynamic	0.466	0.444
PLA/ESO/Nano-CaCO3 (100:3:4 wt%)	Static	0.456	0.420
	Dynamic	0.421	0.400
PLA/ESO/zeolite (100:3:4 wt%)	Static	0.535	0.490
	Dynamic	0.501	0.467

**Table 5 membranes-11-00915-t005:** Moisture permeability of different films by moisture permeable cup (the thickness is from 30 μm to 50 μm).

Film Material	WVTR/(g/m^2^, 24 h), 38 °C, (90–0)%RH
Pure PLA	263.10 ± 13.64
PLA/LLDPE (95:5 wt%)	165.44 ± 8.34
PLA/HDPE (95:5 wt%)	122.12 ± 5.22
PLA/EPDM-g-GMA (95:5 wt%)	88.81 ± 6.43
PLA/ESO/nano-CaCO3(100:3:4 wt%)	65.88 ± 4.36
PLA/ESO/Zeolite (100:3:4 wt%)	67.62 ± 3.54

**Table 6 membranes-11-00915-t006:** Moisture permeability of Al_2_O_3_-deposited BOPLA film by moisture permeability tester (the thickness is about 40 μm).

Deposite Degree/Cycle	WVTR (g/m^2^, 24 h), 38 °C, (90–10)%RH
0	80.040 ± 8.336
100	16.352 ± 2.210
200	0.683 ± 0.086
300	0.617 ± 0.115
400	0.505 ± 0.054
500	0.511 ± 0.093

## Data Availability

Not applicable.

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
