# Peer review of "Comparative Study on Water Vapour Resistance of Poly(lactic acid) Films Prepared by Blending, Filling and Surface Deposit"

_membranes, 2021, doi:10.3390/membranes11120915_

Round 1

Reviewer 1 Report

The first revision of the manuscript by S. Wang et al. on water vapour resistance of PLA films shows considerable improvement over the original version. Howewer, there are still some aspects that need to be reworked:

1) There is still a lack of argumentation on the eco-friendliness of the usage of petro-based filler polymers like PE, PP, and EPDM-g-GMA to modify the otherwise biodegradable PLA biopolymer. Also, the amounts of materials used to generate the various films should be given.

2) Data are still inconsistently presented: No data for friction coefficient is given for the second PLA variant BOPLAR films with thin-film deposited Al2O3 (see Tables 2, 3 and 4). On the other hand, only for this variant the contact angle measurements were presented, but not for the other PLA modifications. 

3) There is almost no discussion of the diff. effects of modifications on the WVTR.

4) The conclusions lack a conclusive statement on which modification meets the aim of the study best, i.e. "the most influencing factors of film's water vapour resistance" (l.15). In addition, the choice of material in the study is questioned by the statement "food contact materials containing nano-materials and plasticizers may cause safety problems …" (l. 362-63), whereas it was written otherwise that "All these three polymers … meet he hygienic requirements for the use of food and drugs." (l. 232)

Minor aspects:

Captures of Fig. 11 and Table 6 should include: Al2O3-deposited BOPLA film.

In Figure 10, not improvement of the toughness of PLA film is shown, but the increase of elongation, while strength is reduced (l. 252). Since no values for toughness are presented, the properties discussed should not mixed.

The manuscript needs moderate english language check.

Reviewer 2 Report

e.g. the accuracy of the determination of mechanical characteristics should be taken into account (e.g., tensile strength xx.xxx MPa, elongation at break xx.x %, see Table 1)

Reviewer 3 Report

The author selected the section of "Membrane processing and engineering". Then, I focused on the following point to reconsider the manuscript revised along with the formed reviewers.

(1) Abbreviation should be defined.

(2) Figure 3 and 4 should be combined and please described the values of x and y in Figure 3.

(3) As for Figures 8~11, please describe the motivation why the measurement of longitudinal elongation at break.

(4) Branching of PE backbone shown in Figure 2 is important. This branching of PE does give an impact on the entanglement of PE with PLLA, which would change the mechanical strength of blending film. Such an entanglement in blending should occur at the case of PLLA with EPDM-g-GMA.  Entanglement of polymers might connect with the matrix effect to relate with the expand process of water vapor from high humidity side to the other side as shown in Figure 13. In this study, the film was prepared by a speed of 35 rpm at 155-185oC, which suggested the possibility that the disentanglement rate > shear rate determined the crystallinity of films (e.g. Mouhamad et al., J. Appl. Phys. 116, 1223513 (2014)). This crystallinity of film might determine the water vapour permeability shown in Figure 13.

(5) Title is a "Comparative study on water vapour resistance of PLA films ...". The author listed the data concerning the tensile strength, longitudinal elongation at break, friction coefficient (static and dynamic), and so on. However, there appeared to be no explanation of relationship between those mechanical properties of (blending, filling, both filling, and plasticizing, depositing Al2O3). Therefore, the reviewer could not understand the mechanistic details how the WVTR value was improved by the preparation method shown in Tables 5 and 6. For example, "nan CaCO3 well dispersed in PLA and as a nucleating agent, the film has higher crystallinity (line3 339-340)" 

Author Response

This manuscript is a resubmission of an earlier submission. The following is a list of the peer review reports and author responses from that submission.

Round 1

Reviewer 1 Report

The manuscript of S. Wang et al. on PLA biopolymer film modifications has several flaws:

  1. The motivation of the study, and especially the materials used remain unclear: PLA is introduced as a biopolymer that has the potential to replace fossil-based plastics in packing industry if the drawbacks such as low barrier properties are overcome. However, as fillers two variants of petro-based PE as well as EPDM-g-GMA were used, contradicting a potential sustainable modification; no info for the choice of these fillers is given. In addition, no info on why two different PLA sources were studied is given; the second variant BOPLAR is only used in thin-film deposition analysis, thus no comparison between the two PLA sources can be made. No info is given on the toxicity of the modifications applied.
  2. Not all referred/discussed results is shown as data, i.e. no method & no data is presented for bi-axial stretched PLA films (l. 299); and the text referring to  Fig. 7 (l. 209) states higher toughness values, but on the Y-axis only tensile strength is given in MPa. On the other hand, not all variants are listed in table 4.
  3. The overall effect of the filler modifications on strength is considerable low. No statistical analysis of the data is given; and no info is given of what kind of error bars are presented.
  4. No info is presented for uniformity of plasma-assisted a.l.d. of Al2O3, but it is stated that "uniformity of ultr-thin PLA surfaces" (l. 271) remain a challenge.

The manuscript needs corrections by a native speaker.

Reviewer 2 Report

1) Check the correctness of the formula in Figure 1

2) The more detailed specification of the polymer components used must be provided (incuding their molecular parameters, e.g. molar mass).

3)  In the chapter "Results and Discussion" , it is necessary to interpret the obtained data in more detail and clearly.

Reviewer 3 Report

(1) As a critical point of the manuscript, the water vapor resistance of PLA films prepared by three kinds of methods was not discussed based on their surface properties. Then, the present version of manuscript gave the impression that each experiment was independently performed and gathered. Therefore, general readers might not be able to get the new strategy to design the polymer film materials with different water vapor resistance, based on the surface properties. In such sense, the reviewer could not judge the Novelty (or Originality) and Significance of Content of the present work. 

(2) One of reasons the reviewer considered like (1) was the low quality of Introduction. Reconstruction of the story line of Introduction should be needed so that the motivation why the authors compared three types of modifications of PLA (polymer blending; polymer filling; and surface coating of films) could be understood for general readers. For examples, drawbacks to be improved, in each type of modifications of PLA films were unlikely to be stated in Introduction (lines 45-86). Examples introduced in polymer blending modification of PLA films appeared to have nothing to do with the experimental system the authors  used in the present study (lines 45-59). Therefore, the reviewer could not understand what the authors wanted to clarify (although the reviewer could, of-course, understand respective data).

(3) Another reason the reviewer considered like (1) was that the authors discussed just the overall trend seen in each Figure and Table. However, the details of each data and whether the result were in agreement with the author's prediction were not discussed (Fig.5-Fig.9, Table 1-5). Furthermore, there was probably no relationship of water vapor resistance with the strength of films, friction coefficient, and contact angle of water droplets.

(4) Also, another reason the reviewer considered like (1) was a lack of discussion regarding the water vapor resistance. Figure 10 demonstrates a so-called dissolution-diffusion model on the gas permeation across the polymer membrane. Was this figure really needed for general readers to understand the results in Tables 4 and 5, because the description based on the model mentioned above was unlikely there? Besides, the authors did not discuss the difference in WVTR between the types of PLA films. In addition, why did the authors respectively use two measurement methods to evaluate WVTR? (Tables 4 and 5) 

Thus, as listed in (2)~(4), the critical point stated in (1) should be improved before a consideration in this journal.

As the minor points, ....

(5) In line 285, "the water vapour Water Vapor Transmissoin Rate (WVTA) by..." is OK as English?

(6) How many experiments were the authors performed to give the error bar for each data?